



# Geochemical zones and environmental gradients for soils from the Central Transantarctic Mountains, Antarctica

Melisa A. Diaz[1,2*], Christopher B. Gardner[1,2], Susan A. Welch[1,2], W. Andrew Jackson[3], Byron J. Adams[4], Diana H. Wall[5], Ian D. Hogg[6,7], Noah Fierer[8], W. Berry Lyons[1,2]

[1]School of Earth Sciences, The Ohio State University, Columbus, OH, USA
[2]Byrd Polar and Climate Research Center, The Ohio State University, Columbus, OH, USA
[3]Department of Civil, Environmental, & Construction Engineering, Texas Tech University, Lubbock, TX, USA
[4]Department of Biology, Evolutionary Ecology Laboratories, and Monte L. Bean Museum, Brigham Young University, Provo, UT, USA
[5]Department of Biology and School of Global Environmental Sustainability, Colorado State University, Fort Collins, CO, USA
[6]Canadian High Arctic Research Station, Polar Knowledge Canada, Cambridge Bay, Nunavut, Canada
[7]School of Science, University of Waikato, Hamilton, New Zealand
[8]Department of Ecology and Evolutionary Biology and Cooperative Institute for Research in Environmental Science, University of Colorado Boulder, Boulder, CO, USA

*Correspondence to:* Melisa A. Diaz (diaz.237@osu.edu)

**Abstract.** Previous studies have established links between biodiversity and soil geochemistry in the McMurdo Dry Valleys, Antarctica, where environmental gradients are important determinants of soil biodiversity. However, these gradients are not well established in the Central Transantarctic Mountains, which are thought to represent some of the least hospitable Antarctic soils. We analyzed 220 samples from 11 ice-free areas along the Shackleton Glacier (~85 °S), a major outlet glacier of the East Antarctic Ice Sheet. We established three zones of distinct geochemical gradients near the head of the glacier (upper), central (middle), and at the mouth (lower). The upper zone had the highest water-soluble salt concentrations with total salt concentrations exceeding 80,000 µg g$^{-1}$, while the lower zone had the lowest water-soluble N:P ratios, suggesting that, in addition to other parameters (such as proximity to water/ice), the lower zone likely represents the most favorable ecological habitats. Given the strong dependence of geochemistry with geographic parameters, we established multiple linear regression and random forest models to predict soil geochemical trends given latitude, longitude, elevation, distance from the coast, distance from the glacier, and soil moisture (variables which can be inferred from remote measurements). Confidence in our model predictions was moderately high, with $R^2$ values for total water-soluble salts, water-soluble N:P, $ClO_4^-$, and $ClO_3^-$ of 0.51, 0.42, 0.40, and 0.28, respectively. These modeling results can be used to predict geochemical gradients and estimate salt concentrations for other Transantarctic Mountain soils, information that can ultimately be used to better predict distributions of soil biota in this remote region.



## 1. Introduction


From an ecological standpoint, the least biologically diverse terrestrial systems are those found in extreme physical
and chemical environments. The abundance and diversity of life in soils is dependent on a number of environmental
parameters, including temperature, precipitation, organic matter content, and nutrient availability (Wall et al., 2012). Hot
deserts are typically viewed as one of the least biologically diverse environments. However, cold deserts can often be even
less diverse (Freckman and Virginia, 1998). Soils in Antarctica typically serve as end-members for low habitat suitability
due to their high salt concentrations, low organic carbon, low soil moisture, and low mean annual temperatures (Courtright et
al., 2001).
In the McMurdo Dry Valleys (MDV), organic matter and salt concentrations influence soil communities, where
soils with higher amounts of organic carbon, lower water-soluble N:P ratios, and lower total water-soluble salt
concentrations generally harbor the greatest biomass and biodiversity (Barrett et al., 2006; Bottos et al., 2020; Caruso et al.,
2019; Magalhães et al., 2012). These Antarctic ecosystems are relatively simple and are the only known soil systems where
nematodes and microarthropods (Collembola, Acari) are at the top of the food chain (Freckman and Virginia, 1998; Hogg
and Wall, 2012). Studies of soils in the MDV and Transantarctic Mountains (TAM) have been key to understanding
ecosystem structure and function in extreme terrestrial environments (e.g. Caruso et al., 2019; Collins et al., 2019; Freckman
and Virginia, 1998).
Biological processes in Antarctic soils are largely dependent on the availability, duration, and proximity of soils to
liquid water (Barrett et al., 2006). Due to the seasonality in freezing and thawing events, liquid water acts as a pulse to the
ecosystem, providing water for organisms, but also wetting surface soils and dissolving soluble salts (Webster-Brown et al.,
2010; Zeglin et al., 2009). Experiments of salt thresholds on Antarctic nematodes found that no individuals survived in
highly saline soils (~2,600 mg L$^{-1}$ TDS) (Nkem et al., 2006). Concentrations of soluble salts exist at these concentrations or
higher for high elevation and inland locations in the TAM (Bockheim, 2008; Lyons et al., 2016). Additionally, studies on
TAM soils have found that increased salt concentrations lead to a decrease in soil biodiversity in older soils compared to
younger soils (Magalhães et al., 2012). Yet, despite these inhospitable conditions (e.g. high salt concentrations and glacial
advance and retreat), some organisms are postulated to have found suitable refugia in TAM soils and persisted in isolation
for millions of years and through glacial cycles (Beet et al., 2016; Stevens et al., 2006; Stevens and Hogg, 2003).
It is generally accepted that habitat suitability for invertebrate species in Antarctic soils is driven by a combination
of geochemical, geographic, and geomorphic variables (Bottos et al., 2020; Courtright et al., 2001; Freckman and Virginia,
1998; Magalhães et al., 2012). Geographic variables, such as elevation, can be measured with advanced mapping tools and
satellite imagery; however, surface exposure ages, soil geochemistry and nutrient content require extensive logistical support
and resource allocation for sample collection and analysis. More efficient estimation tools are needed to aid in our ability to
understand and predict habitat suitability for invertebrates throughout the TAM.



With this study, we determined and evaluated geochemical patterns and gradients of water-soluble ions in soils
collected from 11 ice-free areas along the Shackleton Glacier, Central Transantarctic Mountains (CTAM). Particular
attention was given to total water-soluble salt concentrations, N:P ratios, and $ClO_4^-$ and $ClO_3^-$ concentrations, based on their
influence on biodiversity, as determined in previous studies (e.g. Ball et al., 2018; Barrett et al., 2006b; Courtright et al.,
2001; Dragone et al., 2020; Nkem et al., 2006). The geochemical data were compared to geographic parameters to
understand how the physical environment influences the observed geochemical variability. Our results show that water-
soluble ion concentrations and distributions are driven largely by soil geography and surface exposure age. Finally, we
implemented statistical and machine learning techniques to interpolate and predict the soil geochemistry across the region
using geographic variables. Our multiple linear regression and random forest models show that latitude, longitude, elevation,
distance from the coast, distance from the glacier, and soil moisture (all variables currently or soon to be remotely
measurable using maps and satellites) are moderately effective at estimating spatial patterns in TAM soil geochemistry, with
$R^2$ values as high as 0.87. These data will be particularly useful for ecologists seeking to understand refugia and habitat
suitability in Antarctica and similarly harsh, desert environments.
**2. Study sites**
The Shackleton Glacier (~84.5 to 86.4°S; ~130 km long and ~10 km wide) is a S-N trending outlet glacier of the
East Antarctic Ice Sheet (EAIS) located to the west of the Beardmore Glacier and flows through the Queen Maud Mountains
(CTAM) into the Ross Sea (Fig. 1). The elevations of exposed soils range from ~150 m.a.s.l. to >3,500 m.a.s.l. from the
coast towards the Polar Plateau. Long-term climate data are not yet available, but the Shackleton Glacier region is a polar
desert regime, similar to the Beardmore Glacier region, with average annual temperatures well below freezing and little
precipitation (LaPrade, 1984).

During the Last Glacial Maximum (LGM) and glacial periods throughout the Pleistocene, the size and thickness of
the EAIS has been suggested to be greater than current levels (Golledge et al., 2013; Nakada and Lambeck, 1988; Talarico et
al., 2012; Wilson et al., 2018). Outlet glaciers, such as the Shackleton Glacier, may have had the greatest increases in extent,
especially towards the glacier terminus (Golledge et al., 2012; Golledge and Levy, 2011). The behavior of local alpine and
tributary glaciers is not well-constrained, but these glaciers are also believed to have advanced and retreated over the last two
million years (Diaz et al., 2020a; Jackson et al., 2018). As a result, currently exposed soils were overlain and reworked by
fluctuations of the Shackleton Glacier and other tributary and alpine glaciers in the region. Exposure ages range from the
early Holocene to the Miocene, and generally increase with distance from the coast and distance from the glacier (Balter et
al., 2020; Diaz et al., 2020a).

The soils contain a range of water-soluble salts derived primarily from atmospheric deposition and chemical
weathering (Claridge and Campbell, 1968; Diaz et al., 2020b). The major salts are typically nitrate and sulfate salts,
especially at higher elevations and further inland from the coast of the Ross Sea (Diaz et al., 2020b). The solubilities of the



salts vary, but nitrate salts are highly soluble and their occurrence at high elevation and inland locations suggests that those
soils have maintained persistent arid conditions.
**3. Methods**
3.1. Sample collection and preparation
During the 2017-2018 austral summer, 220 surface soil samples (~top 5 cm) were collected from 11 distinct ice-free
areas (Roberts Massif, Schroeder Hill, Mt. Augustana, Bennett Platform, Mt. Heekin, Thanksgiving Valley, Taylor Nunatak,
Mt. Franke, Mt. Wasko, Nilsen Peak, and Mt. Speed) along the Shackleton Glacier, including a subset of 27 samples
previously analyzed for S, N, and O isotopes in nitrate and sulfate (Diaz et al., 2020b). At each area, we collected samples in
transects (ranging from ~200 m to ~2,000 m in length) to maximize the geochemical variability. Our transects were also
designed to capture the LGM transition, with some soils exposed throughout the LGM and others exposed following glacier
retreat. GPS coordinates and elevations were recorded with each sample and later used to estimate the distance from coast
and distance from the glacier (defined as linear distance from the nearest glacier – Shackleton, tributary, or alpine). Once
collected, the samples were stored and shipped frozen (-20 ℃) to The Ohio State University.
Prior to geochemical analysis, the samples were dried at 50 ℃ for at least 72 hours with the loss in mass attributed
to soil moisture content. The dried soils were leached at a 1:5 soil to DI water ratio, and the leachate was filtered through 0.4
μm Nucleopore membrane filters (Diaz et al., 2018, 2020b; Nkem et al., 2006). Due to the low sediment to water ratio, this
leaching technique only dissolves the more water-soluble salts (Toner et al., 2013). These include salts with $ClO_4^-$, $NO_3^-$, $Cl^-$,
$SO_4^{2-}$, $ClO_3^-$, and $CO_3^{2-}$ + $HCO_3^-$. Process blanks were generated and analyzed to account for any contamination from the
leaching process.
3.2. Analytical analysis of water-soluble anions, cations, and nutrients
The analytical techniques used here are similar to those reported by Diaz et al. (2020b). In brief, the analytes
included anions ($F^-$, $Cl^-$, $Br^-$, and $SO_4^{2-}$) which were measured on a Dionex ICS-2100 ion chromatograph, cations ($K^+$, $Na^+$,
$Ca^{2+}$, $Mg^{2+}$, and $Sr^{2+}$) which were measured on a PerkinElmer Optima 8300 Inductively Coupled Plasma-Optical Emission
Spectrometer (ICP-OES), and nutrients ($NO_3^-$ + $NO_2^-$, $PO_4^{3-}$, $H_4SiO_4$, and $NH_3$) which were measured on a Skalar San++
Automated Wet Chemistry Analyzer at The Ohio State University. Perchlorate ($ClO_4^-$) and chlorate ($ClO_3^-$) were measured
using an ion chromatograph-tandem mass spectrometry technique (IC-MS/MS) at Texas Tech University (Jackson et al.,
2012, 2015). All analytes are reported as listed. Total water-soluble salt concentration was calculated as the sum of all
measured cations and anions. The precision of replicated check standards and samples was typically better than 10% for all
major anions, cations and nutrients, and better than 20% for perchlorate and chlorate. Accuracy was typically better than 5%
for all major anions, cations, and nutrients, as determined by the NIST 1643e external reference standard and the 2015 USGS
interlaboratory calibration standard (M-216), and better than 10% for perchlorate and chlorate, as determined by spike



recoveries. Precision and accuracy for individual analytes are located in Table S1. Detection limits for the analytes have been
previous reported (Diaz et al., 2018; Jackson et al., 2012).
3.3. Data interpolation and machine learning
Inverse distance weighted (IDW) interpolations were performed for Bennett Platform, Thanksgiving Valley, and
Roberts Massif using the Geostatistical Analyst tool in ArcMap 10.3. Since IDW is a deterministic method where unknown
values are predicted based on proximity to known values, we chose those three sites as they had the most defined transects
and relatively higher sample density. The interpolation parameters were constant with a power of 4, maximum neighbors of
15, minimum neighbors of 5, and 4 sectors, and a variable search radius. These parameters were chosen such that they
optimize for the lowest mean absolute error.
Multiple linear regressions were generated for all geochemical analytes, except $H_4SiO_4$ (total of 15 dependent
variables), with latitude, longitude, elevation, distance from the coast, distance from the glacier, and soil moisture as
independent variables using built-in functions in R 3.6.3 (R Core Team, 2020). Random forest regression models were
similarly generated using the randomForest library. The random forest model is a machine learning algorithm that utilizes
supervised learning algorithms to predict values given input predictor variables (Breiman, 2001). Multiple decision trees are
run in parallel with a randomized subset of predictor variables, and the aggregate result of each tree is used to generate a
predicted outcome. Since each tree is generated using a random sample and random predictor variables, the random forest
model is effective at minimizing overfitting and handling outliers (Breiman, 2001).
Machine learning algorithms are widely used in variety of disciplines from finance (Patel et al., 2015) to ecology
(Davidson et al., 2009; Peters et al., 2007; Prasad et al., 2006), for both data prediction (regression) and classification.
Recently, these techniques have been used for Earth Science applications, including geologic mapping (Heung et al., 2014;
Kirkwood et al., 2016), air quality monitoring (Stafoggia et al., 2019), and water contaminant tracing (Tesoriero et al., 2017).
We developed a novel application of machine learning to predict concentrations and gradients of water-soluble salts in
Antarctic soils, given set geographic parameters, similar to the approaches developed for stock market and real estate
predictions (Antipov and Pokryshevskaya, 2012; Patel et al., 2015).
For our random forest models, any sparse missing values in Table S2 were estimated by averaging the geochemistry
of the samples collected immediately before and after in the same transect. Missing values due to concentrations below the
detection limit were input as 0. The new imputed dataset was split into a training set representing 86% of the data (n = 189,
Table S3) and a testing set representing the remaining 14% (n = 31, Table S4). The training dataset was used to generate the
random forest models for each analyte. Each of the models were run with 2000 decision trees (ntree = 2000) to minimize the
mean square errors. The number of random variables used for each node split in the decision trees was set to the
recommended regression default of variables/3 to optimize the model randomness, which in our case was 2 (mtry = 2),



following parameters described previously (Breiman, 2001). The scripts developed for both the multiple linear regression
and random forest models are included in the supplementary materials.
**4. Results**
4.1. Geochemistry of upper, middle, and lower zones

The maximum, minimum, mean, standard deviation and coefficient of variation are reported in Table 1 for the
measured geographic and geochemical data. Concentrations of water-soluble ions span up to five orders of magnitude and
are variable across the region. Elevation, distance from the coast, distance from the glacier, and soil moisture are also
variable and span up to three orders of magnitude. The highest elevation samples (> 2,000 m.a.s.l.) were collected from
Schroeder Hill and the greatest soil moisture content is from Mt. Wasko at 12.3%, with a mean of 2.1% for all samples.

Shackleton Glacier region surface soils can be separated into three zones based on their water-soluble geochemistry:
an upper zone near the Polar Plateau, a middle zone near the center of the glacier, and a lower zone where the glacier flows
into the Ross Sea (Figs. 1; 2). The upper zone samples are characterized by the highest total water-soluble salt
concentrations, with the highest values greater than 80,000 $\mu g\ g^{-1}$ at Schroeder Hill, while the lower zone samples have the
lowest total salt concentrations, with the lowest values near 10 $\mu g\ g^{-1}$ at Mt. Wasko (Fig. 2a-c). The middle zone has
intermediate values. Water-soluble N:P molar ratios generally follow a similar trend (Fig. 2d-f). The lowest N:P ratios are in
the lower zone soils, while the middle and upper zones have more variable values. Concentrations of $ClO_4^-$ and $ClO_3^-$ follow
similar trends as the total salts, with less distinction between middle and upper zones, though most concentrations in the
lower zone are below the detection limit (Fig. 2g-l; Table S2).

Observed trends between the zones appear to be driven, at least partially, by geography. Regressions of total water-
soluble salt concentration, water-soluble N:P ratio, and $ClO_3^-$ concentration with elevation, distance from the coast, and
distance from the glacier are all positive (Fig. 2). The strongest relationships are between total salts and elevation, and $ClO_3^-$
and distance from the coast, with $R^2$ values of 0.26 and 0.24, respectively, and p-values < 0.001 (Fig. 2a;2k). The weakest
relationships are between $ClO_4^-$ and distance from the coast and distance from the glacier, with $R^2$ values of 0.01 (Fig. 2h;
2i). Distance from the glacier varies widely between individual zones with frequent overlaps, but there appears to be a
moderate relationship with N:P ratio and total salts (Fig. 2c; 2f). Overall, total salt concentration has the strongest
relationship with geography and $ClO_4^-$ has the weakest relationships.

Ternary diagrams highlight the specific geochemical gradients within and between the zones. The anion ternary
diagram only includes $SO_4^{2-}$, $NO_3^-$, and $Cl^-$, which are the major water-soluble salts in the region (Claridge and Campbell,
1968; Diaz et al., 2020b). Though carbonate and bicarbonate salts have been identified in both lacustrine sediments and soils
in Antarctica, previously measured concentrations in the Shackleton Glacier region were low, ranging from 0.07 to 2.5%,
and bicarbonate salts were not identified in the highest elevation and furthest inland soils (Claridge and Campbell, 1968;
Diaz et al., 2020b; Lyons et al., 2016). The most abundant anion for the upper zone is $SO_4^{2-}$, which is greater than 99% of the





total anion budget in some Schroeder Hill and Roberts Massif samples, though other locations are dominated by $NO_3^-$ (Fig.
3). The anions are more evenly distributed in the middle zone, though the majority of samples are most abundant in $NO_3^-$ and
$Cl^-$. The lower zone has much lower $SO_4^{2-}$ fractions than the upper zone and the dominant anion is generally $Cl^-$. The cation
distribution is very similar for all three zones (Fig. 3). $Na^+ + K^+$ is the most abundant cation pair representing over 90% of
the total cations for many upper and middle zone samples, while $Ca^{2+}$ is the second most abundant. In general, $Mg^{2+}$ is the
least abundant cation across all sampling locations.
4.2. Statistical geochemical variability

A principal component analysis (PCA) was performed in R (using factoextra (Kassambara and Mundt, 2017) and

built in software libraries) to determine which geochemical variables most strongly differ across the samples. For the PCA,
the first two principal components account for over 50% of the total dataset variability at 44.2% and 11.6%, respectively.
The different zones are correlated with different principal components (Fig. 4). The samples from the middle zone are
positively correlated with PC1 and PC2. In the biplot, they plot in the upper right quadrant with high concentrations of $Cl^-$,
$NO_3^-$, water soluble N:P ratio, and $Ca^{2+}$, with a minor influence from soil moisture and $H_4SiO_4$. The upper zone samples
generally plot along PC1 and are most associated with $Sr^{2+}$, $SO_4^{2-}$, $Mg^{2+}$, $Na^+$, $K^+$, $F^-$, $ClO_4^-$, and $ClO_3^-$. The samples from the
lower, more coastal zone are negatively correlated with PC1 and are distinguished by their higher $PO_4^{3-}$ concentrations. Most
samples from all locations plot within the 95% confidence interval ellipses. However, there are two strong outliers from
Schroeder Hill and Mt. Heekin.

Similar to the PCA, we performed a simple Spearman's rank correlation for the entire dataset to visualize the

statistical dependence between all variables. Since a goal of this study is to relate water-soluble ion concentrations to
geography, we focused on latitude, longitude, distance from the coast, distance from the glacier, and soil moisture. The
strongest correlation coefficients are between $Cl^-$ and latitude, elevation, and distance from the coast, and $Sr^{2+}$ and soil
moisture (Fig. 5). Most other correlations are moderate to weak, though there appear to be notably stronger correlations
between $ClO_3^-$ and latitude and distance from coast, $Ca^{2+}$ and longitude, elevation, and distance from coast, $NO_3^-$ and
latitude, and $SO_4^{2-}$ with distance from glacier. Longitude, elevation, and distance from coast have the greatest number of
strong and moderate correlations with the geochemistry data. Outside of the geographic parameters, $Na^+$ is highly correlated
with total water-soluble salts, likely representative of the high $Na^+ + K^+$ percentages (Fig. 3), and $Sr^{2+}$ is highly correlated
with $K^+$, likely reflecting a common ion source.
4.3. Spatial interpolation and machine learning model performance

The total salt concentrations of individual samples at Bennett Platform produce the most defined interpolation

gradient from the glacier front to further inland compared to Roberts Massif and Thanksgiving Valley (Fig. 6). Bennett
Platform also has the smoothest salt concentration contours suggesting that the interpolation model is the strongest and most
robust at this location. The second strongest interpolation is Thanksgiving Valley. Contrary to the measurements at Bennett



Platform, Thanksgiving Valley has the highest salt concentrations in the center of the valley, with lower concentrations to
both the east and west. The lowest concentration contours are closest to the glacier for both Bennett Platform and
Thanksgiving Valley, which is likely related to glacial history since the soils near the glacier are relatively younger than
those further inland based on meteoric $^{10}$Be data (Diaz et al., 2020a). The interpolation from Roberts Massif does not have a
distinguishable spatial trend.

The multiple linear regression and random forest models vary in their strength for the individual analytes. The
highest $R^2$ value from the linear regression is 0.55 for $Sr^{2+}$, while total water-soluble salts, water-soluble N:P ratio, $ClO_4^-$,
and $ClO_3^-$ have values of 0.37, 0.37, 0.10, and 0.33, respectively (Table 2). The lowest $R^2$ value is for $Cl^-$ at 0.05. The p-
values for nearly all analytes are <<0.001, with $Cl^-$ having the only value above 0.05. The highest out-of-the-bag explained
variance values from the random forest models are for $K^+$ and $Sr^{2+}$ at 62% for both analytes. Values for $NO_3^-$, $PO_4^{3-}$, $ClO_4^-$,
and N:P ratio are negative. The explained variance for total salts is 45% and the variance for $ClO_3^-$ is 43%. We also
evaluated the most important and least important variables in the random forest models based on node purity. The most
important variable for the majority of analytes is elevation, while distance from the glacier is most important for N:P ratio
and latitude for $ClO_3^-$ (Table 2). The least important variable is distance from the coast for every analyte, except $ClO_3^-$ and
$NH_3$, for which distance from the glacier is least important.
**5. Discussion**
5.1. Implications for ecological habitat suitability

By establishing geochemical zones for the Shackleton Glacier region, we can better understand the relationship
between geochemistry and geography, and ultimately biogeography. As stated in the introduction, we focused particularly on
total water-soluble salt concentrations, water-soluble N:P ratios, and $ClO_4^-$ and $ClO_3^-$ concentrations.
5.1.1. Elevation and moisture controls on total water-soluble salt gradients

The elevational trends of total salt concentrations at the Shackleton Glacier are similar to those previously described
in the TAM, where soils from higher elevation sites typically have higher salt concentrations (Bottos et al., 2020; Lyons et
al., 2016; Magalhães et al., 2012). Our results are also consistent with those from Scarrow et al. (2014), who found that salt
concentrations typically decreased with distance from the glacier. Our total water-soluble salt interpolation maps highlight
the spatial variability in Shackleton Glacier region soils (Fig. 6). The most spatially variable location is Robert Massif, which
does not appear to follow local elevational, latitudinal, and/or distance inland gradients. This heterogeneity is not necessarily
due to currently active soil leaching, as the soil moisture values are not drastically different between the samples (Table S2).
Though the variability in cation concentrations is likely due to weathering of tills, scree, and bedrock (Claridge and
Campbell, 1968), recent work on the isotopic composition of water-soluble nitrate and sulfate, the major anions in the upper
zone, suggests a common, atmospheric source (Diaz et al., 2020b).



We argue that the heterogeneity in the total salt concentrations at Roberts Massif (Figs. 2; 6) is probably related to
different and complex wetting history, where seasonal snow patch melt may pool in local depressions, transporting water-
soluble salts from slightly higher elevations and/or from saline wet-patches (Levy et al., 2012). This is demonstrated on a
larger scale at Thanksgiving Valley, a glacially carved valley, where the higher concentrations of salts in the center of the
valley are likely due to the transport of salts from nearby higher elevation slopes during melting events. This is further
evidenced by the presence of two small, closed-basin ponds in the center of the valley, which likely formed from glacial melt
and may have been larger in size in the recent past (Diaz et al., 2019). Similarly, streams and meltwater tracks in the MDV
leach soils and carry salts into closed basin, brackish to hyper-saline lakes, where salts are cryoconcentrated over time
(Lyons et al., 1998). Our results suggest that elevation and wetting history are important contributors to total salt gradients in
the Shackleton Glacier region, as they influence the accumulation of salts and subsequent leaching from soils.
5.1.2. Influence of glacial history on water-soluble N:P ratios
Stoichiometric dependencies have been identified for Antarctic terrestrial organisms, where nutrient concentrations,
in addition to soil aridity, limit ecosystem development (Nkem et al., 2006). Since nitrate is primarily derived from
atmospheric deposition and phosphorus is primarily liberated from minerals by chemical weathering in the CTAM, many
inland and higher elevation soils have accumulated high concentrations of $NO_3^-$, resulting in stoichiometric imbalance with
soluble $PO_4^{3-}$ (Ball et al., 2018; Barrett et al., 2007; Diaz et al., 2020b; Lyons et al., 2016; Nkem et al., 2006). As in the
MDV, younger and coastal soils at lower elevations in the Shackleton Glacier region have the lowest water-soluble N:P
ratios, driven by relatively low concentrations of $NO_3^-$ and high concentrations of $PO_4^{3-}$ due to an increase in moisture
content and chemical weathering (Heindel et al., 2017) (Fig. 2; 4). It is not surprising that life was conspicuous in these soils,
with thick lichen growth on several rocks and the presence of both Collembola and mites at Mt. Speed and Mt. Wasko (Fig.
S1). However, despite overall elevational and latitudinal gradients, some inland locations in the middle and upper zones have
water-soluble N:P ratios near those from the lower zone (Fig. 2).
The interpolation model from Bennett Platform shows that some locations near the glacier have lower total water-
soluble salt concentrations (Fig. 6), similar to soils surveyed in the MDV (Bockheim, 2002). However, the samples near the
glacier at Bennett Platform not only have lower total salt concentrations, they also have lower N:P ratios than samples
collected further inland. This is also the case for the middle zone locations (Fig. 2f). We argue this is due to differences in
glacial history between the locations. Our previous work showed that soils near the glacier are younger than soils further
inland in the Shackleton Glacier region (Diaz et al., 2020a). These soils are shielded from nitrate accumulation during glacial
periods, and the recently exposed rocks likely serve as fresh mineral weathering material for $PO_4^{3-}$ mobilization (Heindel et
al., 2017). Recently exposed and relatively nutrient rich soils might be important refugia for invertebrates. Previous
hypotheses have suggested that organisms may have persisted at higher elevations during glacial periods (Bennett et al.,
2016; Stevens and Hogg, 2003). However, abiotic gradients in the Beardmore Glacier region suggest that higher elevation
soils have salt concentrations that would classify them as unsuitable habitats (Lyons et al., 2016). If few organisms survived



glaciations, the near-glacier, relatively P-rich soils may be important in helping communities recover and restructure post-
glaciation.
5.1.3. High and variable $ClO_4^-$ and $ClO_3^-$ concentrations

Our $ClO_4^-$ and $ClO_3^-$ concentrations include the highest measured in Antarctica to date and are comparable to
concentrations from the Atacama and Mojave Deserts (Jackson et al., 2015). Though not a strong correlation, the highest
elevation samples (upper zone) have the highest $ClO_4^-$ and $ClO_3^-$ concentrations (Fig. 2g; 2j). Similar to $NO_3^-$, $ClO_4^-$ and
$ClO_3^-$ are derived from atmospheric deposition and because of their solubilities, their accumulations are related to wetting
and glacial histories (Jackson et al., 2016, 2015). Therefore, soils which have been exposed for long periods of time and have
not experienced snow or ice melt, such as those from Schroeder Hill and Roberts Massif, are able to accumulate high
concentrations of $ClO_4^-$ and $ClO_3^-$. Interestingly, our $ClO_4^-$ concentrations are lower (maximum of ~1.9 g $L^{-1}$) than the
highest recorded tolerance (1.1M (~130 g $L^{-1}$) $NaClO_4$) for the extremotolerant bacteria *Planococcus halocryophilus*, yet a
recent study shows no detectable biomass for Schroeder Hill samples (Dragone et al., 2020). (Per)chlorates are strong
oxidizers and are well established as toxic, thus the concentrations of $ClO_4^-$ and $ClO_3^-$ might be additional, crucial indicators
of habitat suitability. However, the concentrations are highly heterogenous across our sampled locations (Fig. 2k-l), and
unlike $ClO_3^-$, neither the multiple linear regression nor random forest models were able to adequately capture the variability
in $ClO_4^-$ concentrations (Table 2).
5.2. Machine learning as a tool to predict soil geochemical trends

We sought to evaluate our multiple linear regression and random forest models using a testing dataset from the
Shackleton Glacier region (n = 31) and a second dataset from the Darwin Mountains (~80°S) (n = 10) (Magalhães et al.,
2012). Few published/available TAM dataset include sample GPS coordinates, soil moisture, and water-soluble ion
geochemistry. As stated in Section 3.3, the Shackleton Glacier region test data were not included in the random forest model
generation so we could evaluate our models with an independent dataset. For the Darwin dataset, distance from the glacier,
distance from the coast, and elevation were determined using the Reference Elevation Model of Antarctica (REMA), while
location, soil moisture and geochemistry were retrieved from the literature (Howat et al., 2019; Magalhães et al., 2012). We
evaluated all 15 analytes from the original models with the Shackleton dataset and due to a lack of data, only evaluated 7
analytes from the Darwin soils (Figure 7).

Both the multiple linear regression and random forest model outputs are moderately well-correlated for the
Shackleton dataset, as determined by Pearson correlations between the measured and predicted values (Fig. 7a; Table 3). The
random forest models outperform the linear regression models for nearly every analyte, with the notable exceptions of $F^-$,
$Na^+$, and $NO_3^-$, and nearly all p-values are <0.001. $Mg^{2+}$ is the most accurately predicted, with $R^2$ values of 0.79 and 0.52 for
the random forest and linear regression models, respectively (Fig. 7a). In terms of our analytes of interest regarding habitat
suitability, total salts have the strongest correlation in the random forest model ($R^2$ = 0.51), followed by water-soluble N:P



ratio ($R^2 = 0.42$), $ClO_4^-$ ($R^2 = 0.40$), and $ClO_3^-$ ($R^2 = 0.28$). N:P ratio in particular performs significantly better than the linear
regression model ($R^2 = 0.05$). Mean absolute error (MAE) and root mean squared error (RMSE) values indicate that the
random forest models also have a smaller error compared to the multiple linear regression models (Table 4). MAE values are
lower than RMSE values for both models, indicating the strong presence of outliers in the testing dataset. This is
unsurprising as the standard deviation and coefficient of variation values for the entire dataset are relatively large for all
analytes.

Similar to the model performance in the Shackleton Glacier region, the water-soluble ion predictions for the Darwin

Glacier region are more strongly correlated with measured values in the random forest models compared to the multiple
linear regressions (Fig. 7b). In fact, the linear regression models fail for the Darwin samples and all concentration outputs are
negative, which is likely due to overfitting during model generation. MAE and RSME values for both models are much
higher than those for the Shackleton dataset (Table 4). On the other hand, the random forest models perform particularly well
for some analytes. Though a small sample size, the $R^2$ values for $Mg^{2+}$ and $K^+$ are 0.87, with p-values $<<0.001$. Total salts is
moderately correlated ($R^2 = 0.44$) and N:P ratio has an $R^2$ value of 0.01, indicating poor model performance. It is unclear
why $Mg^{2+}$ and $K^+$ are the most accurately predicted, though we suspect that this is due to weathering trends of local lithology
across the TAM, since chemical weathering is probably the major source of these ions.

It should be noted that the $R^2$ values simply measure the strength of the correlations between the measured and

predicted values. We performed slope tests by fitting bivariate lines using the standardized major axis (SMA) to further
understand the relationship between the two values using the smatr library in R (Warton et al., 2012). For this test, we
specifically evaluated the null hypothesis ($H_0$) where slope = 1, which would indicate whether an ideal, direct 1:1
relationship exists between the measured and predicted values. Test statistic values (t) were used to measure the sample
correlation between the residuals and fitted values (Warton et al., 2012). Test statistic values near 1 indicate that we reject
the null hypothesis. In other words, higher absolute test statistic values indicate a slope other than 1. Of the 15 analytes in the
Shackleton dataset, 7 analytes have slopes near 1 for the multiple linear regression models and 6 for the random forest, as
indicated by test statistic values less than 0.5. For the Darwin, only one analyte, $NO_3^-$, has a test statistic value less than 0.5
(Fig. 7; Table 3).

These data indicate that while some analytes have high correlations between measured and predicted values, the

models perform best with the Shackleton Glacier region soils. Additionally, though the relationship may not be 1:1, the
random forest models are effective at predicting the measured geochemical gradients. For example, similar to our data, the
Darwin Glacier samples generally have greater water-soluble N:P ratios and total water-soluble salt concentrations further
from the glacier and at higher elevations (Magalhães et al., 2012), a trend that is reflected by our model results despite offset
values. Additionally, corrections for the offset of the model from a slope = 1 (i.e. multiplying the model output value by the
regression slope) can be made to better estimate specific concentrations, though the difference between modeled and





measured values can still be up to 2x greater. Our sample size for building the multiple linear regression and random forest
models is small. We anticipate that, as more data are collected throughout the CTAM, these data can be added to the model
training dataset, expanding our prediction capabilities and increasing model reliability.

**6. Conclusions**

The soil ecosystems found in the Transantarctic Mountains are among the least diverse on Earth and their structure
is influenced by environmental factors. We characterized environmental and geochemical gradients in the Shackleton Glacier
region, which aid in our understanding of the abiotic properties in soils governing biodiversity and biogeography. The 220
samples we analyzed represent a wide range of soil environments: those with different elevation, latitude, longitude, glacial
history, and geochemistry. We determined three soil zones: an upper zone near the head of the glacier which is characterized
by high total water-soluble salt concentrations, high water-soluble N:P ratios, and high $ClO_4^-$ and $ClO_3^-$ concentrations, a
lower zone with low total salt concentrations and higher $PO_4^{3-}$ concentrations, and a middle zone with intermediate values.
The zones help elucidate the geographic influences on soil geochemistry. In addition, our total water-soluble salt
interpolations at Roberts Massif, Bennett Platform, and Thanksgiving Valley reflect the local small-scale variability of salt
concentrations and possible influences from soil age and wetting history.
Similar to previous studies, our results suggest that high elevation and inland soils, such as those from the upper
zone, were likely unsuitable candidates for refugia during the Last Glacial Maximum. However, glacial advance and retreat
and climate shifts may leach soils, lowering otherwise toxic total water-soluble salt concentrations and N:P ratios. These
more recently exposed soils may be particularly important in maintaining and reviving contemporary and past biological
communities.
Five geographic variables (latitude, longitude, elevation, distance from the coast, and distance from the glacier) and
soil moisture were correlated with soil geochemistry. We used these variables to develop multiple linear regression and
random forest models to predict ion concentrations and geochemical gradients. The model results generally reflected the
measured geochemical variability across the region. Test datasets from the Shackleton and Darwin Glacier regions showed
that the random forest models typically outperformed the multiple linear regression models when correlating measured and
predicted values, especially for the Darwin region. Though most correlations did not exhibit a 1:1 relationship and had
varying slopes, the random forest models were able to adequately predict geochemical gradients, as demonstrated by
moderate to high $R^2$ values between measured and model predicted concentrations. As terrestrial Antarctic geochemical
databases expand and are included in the random forest model training dataset, we anticipate the model's predictive
capabilities will expand and improve as well. While these results are currently most applicable for Central Transantarctic
Mountain soils, similar techniques can be applied to other hyper-arid environments (e.g. Namib and Atacama Deserts, Mars)
to inform patterns of biodiversity and biogeography.





**Author Contributions**

The project was designed and funded by BJA, DHW, IDH, NF, and WBL. Fieldwork was conducted by BJA, DHW, IDH, NF, and MAD. CBG, SAW, and MAD prepared and analyzed the samples for water-soluble ion and nutrient analyses. WAJ prepared and analyzed the samples for $ClO_4^-$ and $ClO_3^-$. MAD generated the scripts and performed the analyses for the IDW interpolations, multiple linear regression, and random forest models. MAD wrote the article with contributions and edits from all authors.

**Data Availability Statement**

The datasets generated for this study are included in the article or supplementary materials.

**Competing Interests**

The authors declare that they have no conflict of interest.

**Acknowledgments**

We thank the United States Antarctic Program (USAP), Antarctic Science Contractors (ASC), Petroleum Helicopters Inc. (PHI), and Marci Shaver-Adams for logistical and field support. Additionally, we thank Daniel Gilbert for help with initial laboratory analyses at The Ohio State University. This work was supported by NSF OPP grants 1341631 (WBL), 1341618 (DHW), 1341629 (NF), 1341736 (BJA), and NSF GRFP fellowship 60041697 (MAD). Geospatial support for this work provided by the Polar Geospatial Center under NSF OPP grants 1043681 and 1559691.



**Biogeosciences** Open Access
Discussions
EGU

**Figures**

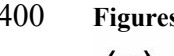


Figure 1. Samples were collected and analyzed from the exposed soils along the Shackleton Glacier, a major outlet glacier of
the EAIS (a), in three zones. The upper zone (b) was located at the head of Shackleton Glacier, the middle zone (c) was the
central portion, and the lower zone (d) was at the mouth of the glacier where it drains into the Ross Sea. Satellite images
were provided courtesy of the Polar Geospatial Center (PGC).



Figure 2. Total water-soluble salts, water-soluble N:P molar ratio, and $ClO_4^-$ and $ClO_3^-$ concentrations (log scale) were compared to elevation, distance from the coast, and distance from the glacier for samples from the three geographic zones. Linear regression lines are plotted and $R^2$ values are reported for each relationship. The horizontal orange line represents nematode salt tolerance of ~2,600 (Nkem et al., 2006) and the green line represents the Redfield ratio, N:P = 16 for phytoplankton in the ocean.

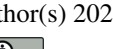




Figure 3. Anion and cation ternary diagrams for the three geographic zones.





Figure 4. Principal component analysis (PCA) biplot generated in R using factoextra and built in software libraries with all
anions, cations, nutrients, and soil moisture for the three geographic zones. Principal component 1 and principal component 2
are plotted on the x and y axes, respectively. Shaded ellipses represent 95% confidence intervals.



Figure 5. Spearman's rank correlation matrix generated in R using the corrplot library. The colors represent correlation coefficients, indicating the strength and magnitude of the correlation. The blue box indicates the geographic variables and soil moisture, which were variables used in the multiple linear regression and random forest models.





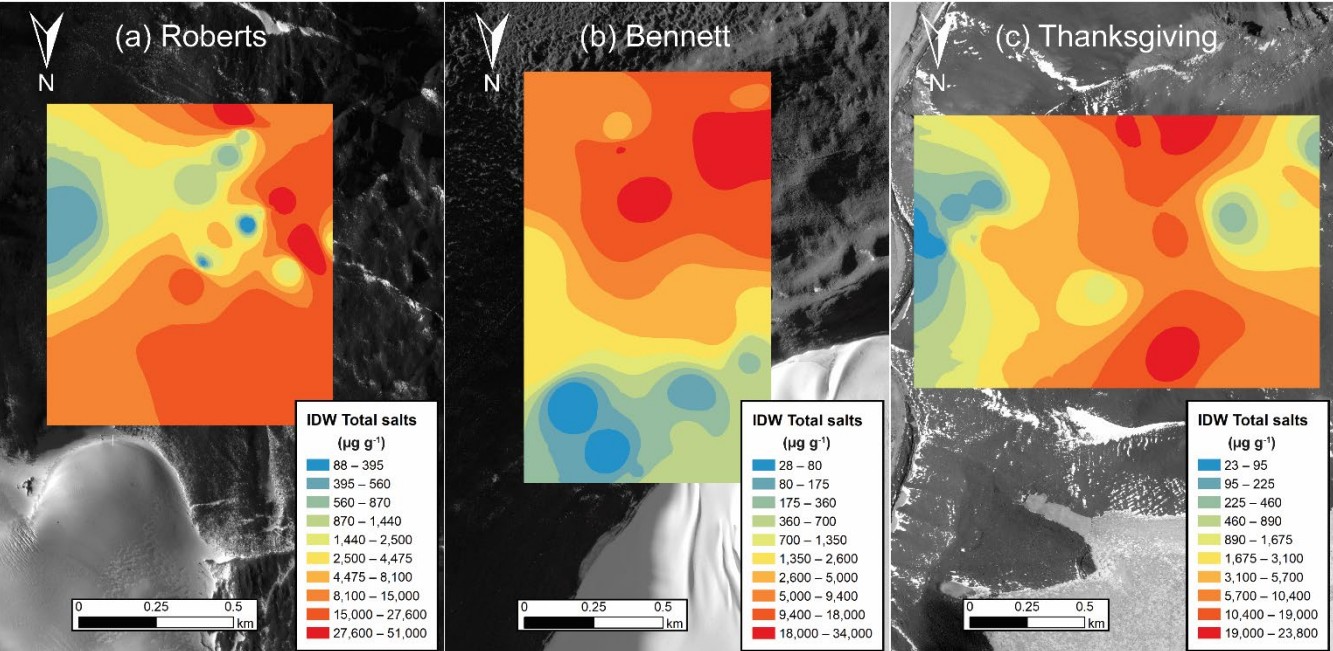

428

Figure 6. Inverse distance weighted (IDW) interpolations of total salt concentration for Roberts Massif (a), Bennett Platform
(b), and Thanksgiving Valley (c). The color scale represents the 10 natural breaks in the data. Interpolations were created and
mapped using the Geostatistical Analyst tool in ArcMap 10.3.





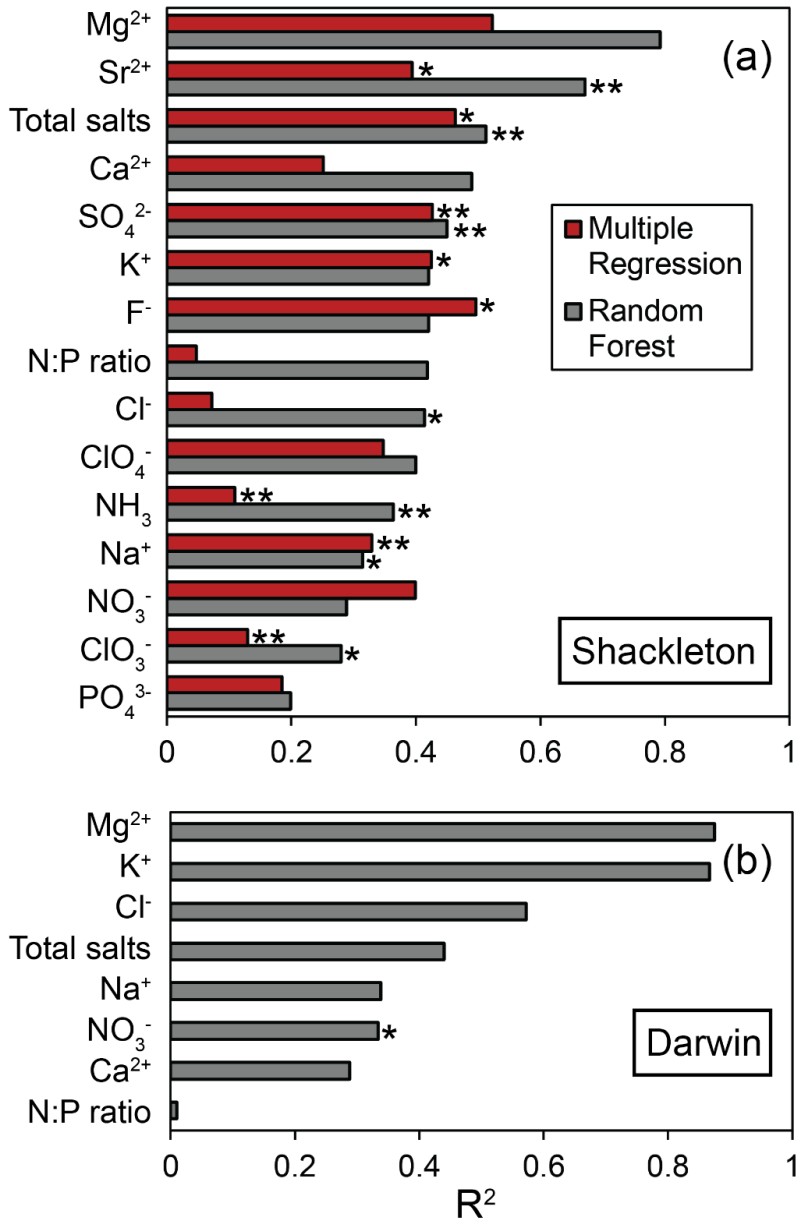

Figure 7. $R^2$ values for the multiple linear regression and random forest model predicted and measured values for the different analytes (Table 3). Test datasets include the Shackleton Glacier region (n =31) and the Darwin Glacier region (n = 10) (Magalhães et al., 2012). Analytes with slopes near 1, indicating good agreement between measured and predicted values, are indicated (* t < 0.5; ** t < 0.20).





Table 1. Overview of geography, soil moisture, and water-soluble ions from the Shackleton Glacier region. The minimum
values reported are those within the detection limits. Individual sample concentrations are detailed in Table S2.

| | Max | Min | Mean | STD | CV |
|---|---|---|---|---|---|
| Elevation (m) | 2,220 | 150 | 1,130 | 551 | 48 |
| Distance from coast (km) | 120 | 1 | 55 | 38 | 68 |
| Distance from glacier (m) | 1,940 | 1 | 519 | 472 | 90 |
| Soil moisture (%) | 12.3 | 0.1 | 2.1 | 2.1 | 102 |
| $F^-$ ($\mu g\ g^{-1}$) | 120 | 0.39 | 8.87 | 11.78 | 133 |
| $Cl^-$ ($\mu g\ g^{-1}$) | 13,600 | 1.59 | 615 | 1,780 | 289 |
| $NO_3^-$ ($\mu g\ g^{-1}$) | 38,400 | 0.10 | 1,470 | 3,450 | 235 |
| $SO_4^{2-}$ ($\mu g\ g^{-1}$) | 55,300 | 0.08 | 4,390 | 8,080 | 184 |
| $PO_4^{3-}$ ($\mu g\ kg^{-1}$) | 4,200 | 76.09 | 381 | 560 | 147 |
| $ClO_4^-$ ($\mu g\ kg^{-1}$) | 75,000 | 0.35 | 985 | 6,020 | 611 |
| $ClO_3^-$ ($\mu g\ kg^{-1}$) | 14,500 | 1.00 | 1,170 | 2,500 | 214 |
| $Ca^{2+}$ ($\mu g\ g^{-1}$) | 4,400 | 0.55 | 839 | 1,160 | 139 |
| $Mg^{2+}$ ($\mu g\ g^{-1}$) | 6,280 | 0.12 | 293 | 705 | 240 |
| $Na^+$ ($\mu g\ g^{-1}$) | 25,300 | 0.39 | 1,140 | 2,880 | 252 |
| $K^+$ ($\mu g\ g^{-1}$) | 440 | 0.86 | 28.31 | 51.61 | 182 |
| $Sr^{2+}$ ($\mu g\ g^{-1}$) | 46.61 | 0.01 | 8.63 | 10.31 | 119 |
| $H_4SiO_4$ ($\mu g\ g^{-1}$) | 60.78 | 1.14 | 21.78 | 11.03 | 50.67 |
| $NH_3$ ($\mu g\ kg^{-1}$) | 5,080 | 18.85 | 324 | 587 | 181 |
| N:P ratio (molar) | 526,000 | 0.29 | 23,600 | 62,700 | 266 |
| Total salt ($\mu g\ g^{-1}$) | 80,500 | 9.46 | 7,932 | 13,300 | 167 |
| STD, standard deviation; CV, coefficient of variation | | | | | |






Table 2. Out-of-the-bag multiple linear regression and random forest model statistics generated in R.

| | Multiple regression | | Random forest | | |
| --- | --- | --- | --- | --- | --- |
| | $R^2$ | p-value | Variance explained (%) | Most important variable | Least important variable |
| $F^-$ | 0.27 | <<0.001 | 36 | Elevation | Distance from coast |
| $Cl^-$ | 0.05 | 0.082 | 20 | Elevation | Distance from coast |
| $NO_3^-$ | 0.18 | <<0.001 | -4 | Distance from glacier | Distance from coast |
| $SO_4^{2-}$ | 0.37 | <<0.001 | 44 | Elevation | Distance from coast |
| $PO_4^{3-}$ | 0.16 | 0.017 | -7 | Latitude | Distance from coast |
| $ClO_4^-$ | 0.1 | 0.010 | -3 | Elevation | Distance from coast |
| $ClO_3^-$ | 0.33 | <<0.001 | 43 | Latitude | Distance from glacier |
| $Ca^{2+}$ | 0.26 | <<0.001 | 46 | Soil moisture | Distance from coast |
| $Mg^{2+}$ | 0.29 | <<0.001 | 22 | Elevation | Distance from coast |
| $Na^+$ | 0.21 | <<0.001 | 38 | Elevation | Distance from coast |
| $K^+$ | 0.4 | <<0.001 | 62 | Elevation | Distance from coast |
| $Sr^{2+}$ | 0.55 | <<0.001 | 62 | Elevation | Distance from coast |
| $NH_3$ | 0.29 | <<0.001 | 54 | Elevation | Distance from glacier |
| N:P | 0.37 | <<0.001 | -3 | Distance from glacier | Distance from coast |
| Total salts | 0.37 | <<0.001 | 45 | Elevation | Distance from coast |





Table 3. Multiple linear regression and random forest statistics between predicted and measured concentrations from the
Shackleton and Darwin Glacier regions. $R^2$ and p-values are reported for the correlations between measured and predicted
concentrations. Regression slopes and test statistic values (t) were calculated using the smatr library (Warton et al., 2012) in
R to evaluate the null hypothesis ($H_0$) of slope = 1. Higher test statistic values (closer to one) indicate that we reject the null
hypothesis.

| Analyte | Multiple Linear Regression | | | | Random Forest | | | |
|---|---|---|---|---|---|---|---|---|
| | $R^2$ | p-value | Reg. slope | Test statistic (t) for $H_0$ slope = 1 | $R^2$ | p-value | Reg. slope | Test statistic (t) for $H_0$ slope = 1 |
| **Shackleton** | | | | | | | | |
| $Mg^{2+}$ | 0.52 | <<0.001 | 0.52 | -0.711 | 0.79 | <<0.001 | 0.58 | 0.780 |
| $Sr^{2+}$ | 0.39 | <0.001 | 1.22 | 0.247* | 0.67 | <<0.001 | 0.91 | -0.166** |
| Total salts | 0.46 | <<0.001 | 0.76 | -0.343* | 0.51 | <<0.001 | 0.93 | -0.107** |
| $Ca^{2+}$ | 0.25 | 0.004 | 0.42 | -0.747 | 0.49 | <<0.001 | 0.61 | -0.586 |
| $SO_4^{2-}$ | 0.43 | <<0.001 | 1.07 | 0.093** | 0.45 | <<0.001 | 1.10 | 0.130** |
| $K^+$ | 0.42 | <<0.001 | 1.54 | 0.504* | 0.42 | <<0.001 | 1.79 | 0.629 |
| $F^-$ | 0.50 | <<0.001 | 1.22 | 0.267* | 0.42 | <0.001 | 1.78 | 0.617 |
| N:P ratio | 0.05 | 0.241 | 0.59 | -0.517 | 0.42 | <<0.001 | 0.35 | -0.867 |
| $Cl^-$ | 0.07 | 0.144 | 0.28 | -0.867 | 0.41 | <<0.001 | 0.70 | -0.424* |
| $ClO_4^-$ | 0.35 | <0.001 | 2.01 | 0.685 | 0.40 | <0.001 | 3.40 | 0.897 |
| $NH_3$ | 0.11 | 0.070 | 1.04 | 0.037** | 0.36 | <0.001 | 1.09 | 0.106** |
| $Na^+$ | 0.33 | <0.001 | 0.91 | -0.112** | 0.31 | 0.001 | 1.54 | 0.473* |
| $NO_3^-$ | 0.40 | <0.001 | 0.47 | -0.725 | 0.29 | 0.002 | 0.56 | -0.594 |
| $ClO_3^-$ | 0.13 | 0.043 | 1.20 | 0.197** | 0.28 | 0.002 | 0.71 | -0.382* |
| $PO_4^{3-}$ | 0.18 | 0.016 | 0.50 | -0.645 | 0.20 | 0.022 | 0.15 | -0.967 |
| **Darwin** | | | | | | | | |
| $Mg^{2+}$ | - | - | - | - | 0.87 | <<0.001 | 0.39 | -0.948 |
| $K^+$ | - | - | - | - | 0.87 | <<0.001 | 0.49 | -0.895 |
| $Cl^-$ | - | - | - | - | 0.57 | 0.011 | 0.13 | -0.984 |
| Total salts | - | - | - | - | 0.44 | 0.001 | 3.25 | 0.940 |
| $Na^+$ | - | - | - | - | 0.34 | 0.078 | 0.23 | -0.931 |
| $NO_3^-$ | - | - | - | - | 0.33 | 0.080 | 0.65 | -0.476* |
| $Ca^{2+}$ | - | - | - | - | 0.29 | 0.110 | 0.17 | -0.961 |





| N:P ratio | - | - | - | - | 0.01 | 0.765 | 8.04 | 0.970 |
|---|---|---|---|---|---|---|---|---|
| * t < 0.5; ** t < 0.20 | | | | | | | | |






Table 4. Multiple linear regression and random forest model mean absolute error (MAE) and root mean square error (RMSE)

| Analyte | Multiple Linear Regression | | Random Forest | |
|---|---|---|---|---|
| | MAE | RMSE | MAE | RMSE |
| Shackleton | | | | |
| $Mg^{2+}$ | 300 | 461 | 204 | 347 |
| $Sr^{2+}$ | 3.74 | 4.96 | 1.83 | 2.90 |
| Total salts | 5,640 | 7,070 | 4,400 | 7,030 |
| $Ca^{2+}$ | 797 | 1,100 | 554 | 912 |
| $SO_4^{2-}$ | 3,310 | 3,890 | 2,200 | 3,780 |
| $K^+$ | 15.86 | 21.16 | 13.48 | 25.61 |
| $F^-$ | 3.14 | 4.19 | 3.13 | 6.31 |
| N:P ratio | 39,700 | 59,300 | 7,310 | 17,210 |
| $Cl^-$ | 936 | 1,540 | 658 | 1,240 |
| $ClO_4^-$ | 1,180 | 1,560 | 875 | 2,960 |
| $NH_3$ | 214 | 301 | 158 | 244 |
| $Na^+$ | 883 | 1,170 | 918 | 1,730 |
| $NO_3^-$ | 1,200 | 1,910 | 1,130 | 2,040 |
| $ClO_3^-$ | 1,110 | 1,630 | 343 | 1,050 |
| $PO_4^{3-}$ | 428 | 690 | 261 | 742 |
| Darwin | | | | |
| $Mg^{2+}$ | 6,300 | 6,320 | 302 | 475 |
| $K^+$ | 1,060 | 1,060 | 13.33 | 15.84 |
| $Cl^-$ | 206,000 | 206,000 | 2,140 | 3,330 |
| Total salts | 215,000 | 215,000 | 5,540 | 7,590 |
| $Na^+$ | 8,330 | 8,530 | 1,500 | 2,600 |
| $NO_3^-$ | 128,000 | 128,000 | 3,260 | 4,870 |
| $Ca^{2+}$ | 70,300 | 70,300 | 1,410 | 2,070 |
| N:P ratio | 18,100,000 | 18,100,000 | 18,700 | 46,900 |
| MAE, mean absolute error; RMSE, root mean squared error | | | | |






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
