# Peer review of "Central Transantarctic Mountains, Antarctica"

_Biogeosciences, 2020_

## Referee Comment (RC2) · Natasja van Gestel (Referee) · 4 Nov 2020

Review of "Geochemical zones and environmental gradients for soils from the Central Transantarctic Mountains, Antarctica"

This paper is very well written, and it is very interesting. The focus is on predicting spatial patterns of water-soluble salts and the ratio of N to P within 11 distinct ice-free sites along a glacier in Antarctica. Ultimately, the models that best predict those patterns could help find refugia of soil invertebrates who may be sensitive to high salt concentrations. I applaud the incorporation of the data and the R code, so that this research is reproducible.

[Figure]

My major concerns, which are easily addressed, are the statistics. I only devote so much time of describing these in detail, because I found it a most interesting paper that I believe should be published. But the statistical approaches should be sound and match the experimental design and follow the assumptions of linear models (otherwise model parameters cannot be properly interpreted). I would be happy to provide more guidance if needed.

1) Log-transformation need to be done where necessary. First, the authors have not checked their regression models to see how the residuals show a pattern with fitted values. This is important to do as one of the (several) assumptions in a linear model are that residuals are normally distributed and their spread around the regression line should be the same irrespective of the value of x. I highly suggest that the authors use log-transformed values in the data where needed and do some model checking with the plot() function and other model checking procedures. This will certainly help with that aspect. I was surprised to see that in the figure they did use log-values, but then did not use it in their regression. Needless to say, this this also needs to be done in their random forest models. The random forest model they used explained 43% of the variance in total salts. By using log-transformed values that went up to 75%. This also altered the importance ranking of the variables. Irrespective, elevation remained important (at least for Total Salts, I did not check the others), but others switched. 2) Data, such as NP, can have many zeroes. For example, the NP training data set had 116 zeroes of the 189 values. That means that a gaussian distribution of the data set is not followed. Having 0's means also a log-transformation will lead to -Inf, and cannot be used in a model. Solution: consider other family of distributions by using "glm" instead of lm. The "g" stands for generalized, and can thus handle other kinds of distributions. 3) Given that multiple samples were collected at each transect at 11 different locations, some in closer proximity than others, that error structure is not taken into account. For example, two samples from the same transect will likely be more closely related than two samples from different transect. To incorporate this, I propose using a mixed-effects model approach to take into account that multiple samples were obtained

from the same transect. Otherwise: your power is inflated, because your samples are not truly independent from each other (another important assumption in linear models with only fixed effects). 4) For the testing and training data set: rather than randomly sample all 220 observations, randomly sample a proportion of the total transects, say 8 out of 11 transect. Then test it on the remaining 3. That takes into account that the observations within a transect are not entirely independent. 5) For PCA: this is another linear model. I found no information on whether the authors performed any visualization to see if the patterns are linear. Given that the data show non-normal distributions, please do revisit this. Also: is the PCA based on the covariance or the correlation matrix? So, information is missing. 6) Lastly, and I refer to Figure 2: the panels are great, but it also highlights that the authors looked at every possible relationship of the total water-soluble salts, N:P, and ClO4- and ClO3- concentrations. However: the more comparisons are made, the higher the probability of making a type 1 error, unless you make the alpha more stringent. In a scenario like this I would recommend something like a Bonferroni correction.

Minor concerns: Replace 'environmental parameters' with 'environmental variables' or 'environmental conditions'. From a model-perspective, parameters are associated with models, e.g. coefficients are parameters. Variables are the data.

L. 158 It is mean squared error.

Figure 2: • Please add the meaning of the blue, yellow and gray colors. It is evident from the next figure, but having it already here will help the reader. Figures are standalone and should be interpretable without having to look for info elsewhere in the paper.

• Also, technically: if a relationship is not significant, one should not show the best-fit line. However, rather than removing them, I would suggest adding dashed lines instead for those where P-values are greater than 0.05.

Supplementary information: Please add the "library(readxl)" to the R script. Otherwise

users will get an error message: the read_excel() function is used, which is from that package.
* * *

---

## Author Comment (AC1) · 5 Dec 2020

Dear Dr. van Groenigen,

We are pleased to submit our revised manuscript, "Geochemical zones and environmental gradients for soils from the Central Transantarctic Mountains, Antarctica". We are very thankful to both Dr. Convey and Dr. van Gestel for providing thoughtful comments, suggestions, and edits, which will guide our revision. In particular, we thank Dr. Convey for helping to improve the interdisciplinary value of this work and Dr. van Gestel for helping to improve the statistical significance. We have responded to the reviewer comments below, which serve as the outline for our revision. The most significant change was log transforming the geochemical data, which resulted in Gaussian distributions for both the values and model residuals. This does not change our conclusions and instead strengthens our results and models. We thank the reviewers once again.

Best regards, Melisa Diaz (on behalf of all authors)

——Referee 1: Peter Convey (Referee) pcon@bas.ac.uk

1) L45-6: as written, the sentence unintentionally I think implies these ecosystems might be restricted to the Dry Valleys. More accurately, systems containing only microarthropods and microinvertebrates are known throughout Antarctica, and in the Antarctic Peninsula/Scotia Arc actually extend up to the South Orkney and South Sandwich Islands (60 and 55-59 deg south, respectively), where there are no native true insects (two species of which do occur in parts of the Antarctic Peninsula and South Shetlands). Admittedly these communities are more diverse in species numbers and with greater biomass than those of the Dry Valleys.

-We thank Dr. Convey for catching this error. We will correct this sentence.

2) To expand the 'most extreme environment' literature beyond the Dry Valleys, there is a recent publication by Collins et al. (2020, PNAS) on part of the Transantarctic Mountains, Hodgson et al. 2010 (Antarct Sci) on the Dufek Massif (far eastern Transantarctic Mountains), and Convey & McInnes 2005 (Ecology) on nunataks in Ellworth Land, with the latter two reporting invertebrate communities in which even nematodes are absent.

-We thank Dr. Convey for these references and will be sure to include them.

3) L54: repeat of word 'concentrations' We will make this correction.

4) L59: the two Collins et al papers would be appropriate to cite here, as the most recent and detailed examples of such studies.

-We will add these references.

5) L61: I would add water availability here too

-We will make this correction.

6) L155-6: my ignorance, but what drives the choice of the proportions of samples included in the training and testing sets?

-The sizes for the training and test datasets are chosen to maximize randomness and variability. Ideal training datasets are around 80% of the total dataset for small datasets (as noted by Breiman, 2001). We will add a sentence about this near line 155.

7) L232: is 'out of the bag' a widely used phrase?

-This was a typo and meant to say out-of-the-box. We will make this correction.

8) L299: did the study referred to here use any culture, molecular or eDNA approaches to assess whether any evidence for biota being present?

-This study used a series of cultivation-dependent, cultivation-independent, and metabolic assays. In fact, 20% of the soils had no amplifiable microbial DNA. More information can be found here: https://doi.org/10.1101/2020.08.03.234583.

9) Section 5.2: is there a case that much of the text in this section might be more appropriate as a subsection in Results?

-We agree with Dr. Convey that some of this section would typical be considered results. However, we argue that it is most logical to keep the text in the discussion given the greater context of our findings. In Section 5.2, we discuss the strength of the model and how the strength relates to its predictive capabilities, both for the Shackleton Glacier region and other portions of the Transantarctic Mountains. We believe that this is best suited for the discussion especially since this section builds from Section 4.3, where we evaluate the model performance.

10) L366: the description of this process whereby 'refuges' are effectively mobile, moving as the glacier front/edge expands or contracts, reminds me of the suggestion of 'temporal refugia' that has been made in the entirely different situation of areas containing multiple but individually short-lived geothermal refugia (Convey & Smith 2006 J Veg Sci; Fraser et al. 2014 PNAS).

-We thank Dr. Convey for suggesting this literature and will incorporate it. This could certainly be considered "temporal refugia".

——Referee 2: Natasja van Gestel (Referee) natasja.van-gestel@ttu.edu

1) Log-transformation need to be done where necessary. First, the authors have not checked their regression models to see how the residuals show a pattern with fitted values. This is important to do as one of the (several) assumptions in a linear model are that residuals are normally distributed and their spread around the regression line should be the same irrespective of the value of x. I highly suggest that the authors use log-transformed values in the data where needed and do some model checking with the plot() function and other model checking procedures. This will certainly help with that aspect. I was surprised to see that in the figure they did use log-values, but then did not use it in their regression. Needless to say, this this also needs to be done in their random forest models. The random forest model they used explained 43% of the variance in total salts. By using log-transformed values that went up to 75%. This also altered the importance ranking of the variables. Irrespective, elevation remained important (at least for Total Salts, I did not check the others), but others switched.

-We thank Dr. van Gestel for pointing out this assumption in our statistical methods. We have log transformed our geochemical data in the regressions and have tested the normality. All regressions have a normal distribution as tested by plotting the residuals and running a Jarque-Bara Test. We will indicate this in the text, update our figures, and add the analyses to our code.

2) Data, such as NP, can have many zeroes. For example, the NP training data set

had 116 zeroes of the 189 values. That means that a gaussian distribution of the data set is not followed. Having 0's means also a log-transformation will lead to -Inf, and cannot be used in a model. Solution: consider other family of distributions by using "glm" instead of lm. The "g" stands for generalized, and can thus handle other kinds of distributions.

-We have chosen to simply remove the zeroes and replaced them with blank (NA) values since we are log transforming the data. As such, we have also added na.rm functions to our models to exclude the NA values. We tested with both glm and lm and the results/strength are the same. Additionally, by log transforming the concentration data, we have achieved a normal distribution. Therefore, we have continued to use the linear model, but thank Dr. van Gestel for the suggestion on generalized linear models and will consider the model in future studies.

3) Given that multiple samples were collected at each transect at 11 different locations, some in closer proximity than others, that error structure is not taken into account. For example, two samples from the same transect will likely be more closely related than two samples from different transect. To incorporate this, I propose using a mixed-effects model approach to take into account that multiple samples were obtained from the same transect. Otherwise: your power is inflated, because your samples are not truly independent from each other (another important assumption in linear models with only fixed effects).

-We thank Dr. van Gestel for the suggestion on using a mixed model approach. From our understanding, a mixed model approach typically would best be applied in a situation where random plus fixed effects are not considered, such as proximity to other samples. However, we argue that by performing our regressions with geography (distance from glacier, elevation latitude, etc.), we have in fact accounted for random effects. Additionally, we have further researched mixed effects models. In summary, by performing the linear regressions, we are ultimately solving for least squares, which provides the best linear unbiased estimator. This is essentially the Gauss–Markov theorem. By solving for least squares, we have inherently proven unbiased estimators and only consider fixed effects. We thank Dr. van Gestel for helping to lead us on this informative endeavor.

4) For the testing and training data set: rather than randomly sample all 220 observations, randomly sample a proportion of the total transects, say 8 out of 11 transect. Then test it on the remaining 3. That takes into account that the observations within a transect are not entirely independent.

-An important point to make here and above (#3) is that the samples are not necessarily related because of the transect sampling scheme. Any partial dependence is fundamentally driven by latitude and longitude, which are considerations in our machine learning models. This is especially important because we are testing individual points and not transects. We have also chosen to randomly sample all 220 observations to further randomize our bias and increase potential sample value variability within the datasets. As Dr. van Gestel points out, we have about 3 transects with about 8 samples each from 11 locations. For simplicity, let's assume that this corresponds to 30 transects. In order to keep the number of observation in the training and test datasets to ∼85% and 15%, respectively, we would only be able to have about 4 transects in the training set. Considering that our ion concentrations span many orders of magnitude throughout the region (see Table 1), it is unlikely we are going to capture enough randomness and variance to generate strong enough models for outliers using a transect approach. This is entirely due to sample size, which we repeatedly recognize as a weakness of this study. We will certainly consider the approach proposed by Dr. van Gestel in the future when the number of observations is much greater.

5) For PCA: this is another linear model. I found no information on whether the authors performed any visualization to see if the patterns are linear. Given that the data show non-normal distributions, please do revisit this. Also: is the PCA based on the covariance or the correlation matrix? So, information is missing.

-We deflect to the points we have made earlier in this review regarding gaussian distributions with our log transformed data. We will add text regarding our observations in Section 4. We apologize for not specifying the PCA. We will update the figure and PCA text to reflect the transformations we have applied to the data and explicitly state that the PCA is based on covariance.

6) Lastly, and I refer to Figure 2: the panels are great, but it also highlights that the authors looked at every possible relationship of the total water-soluble salts, N:P, and ClO4- and ClO3- concentrations. However: the more comparisons are made, the higher the probability of making a type 1 error, unless you make the alpha more stringent. In a scenario like this I would recommend something like a Bonferroni correction.

-We thank Dr. van Gestel for suggesting the Bonferroni correction. We have applied the correction, which only affected the relationship between ClO3- and elevation. We will update the figure accordingly.

7) Minor concerns: Replace 'environmental parameters' with 'environmental variables' or 'environmental conditions'. From a model-perspective, parameters are associated with models, e.g. coefficients are parameters. Variables are the data.

-We will make this correction.

8) L. 158 It is mean squared error.

-We will make this correction.

9) Figure 2: Please add the meaning of the blue, yellow and gray colors. It is evident from the next figure, but having it already here will help the reader. Figures are standalone and should be interpretable without having to look for info elsewhere in the paper. Also, technically: if a relationship is not significant, one should not show the best-fit line. However, rather than removing them, I would suggest adding dashed lines instead for those where P-values are greater than 0.05.

-We will add the zone coloring to the figure caption. We will also follow Dr. van Gestel's

suggestion and add dashed lines for non-significant relationships with the Bonferonni correction applied (only changes for ClO3- vs. elevation).

10) Supplementary information: Please add the "library(readxl)" to the R script. Otherwise users will get an error message: the read_excel() function is used, which is from that package.

-We will add this to the script and apologize for the error.

---

## Author Response (AR1)

Dear Dr. van Groenigen,

We are pleased to submit our revised manuscript, "Geochemical zones and environmental gradients for soils from the Central Transantarctic Mountains, Antarctica". We have incorporated the suggestions from Dr. Convey and Dr. van Gestel and updated the text and figures accordingly. As stated previously, we thank Dr. Convey for helping to improve the interdisciplinary value of this work and Dr. van Gestel for helping to improve the statistical significance.

The changes we have made to the manuscript are recorded in-line in the "tracked changes" file. We have also included the line numbers in our responses below. The most significant change was log transforming the geochemical data, which resulted in Gaussian distributions for both the values and model residuals. As such, we have updated figures 2&7, tables 2-4, and the associated text. This edit did not change our conclusions and instead strengthened our results and models. We hope that with all reviewer comments addressed, our revised manuscript is acceptable for publication.

Best regards,

Melisa Diaz (on behalf of all authors)

Postdoctoral Scholar
Woods Hole Oceanographic Institution
The Ohio State University
Byrd Polar and Climate Research Center

*Peter Convey (Referee) pcon@bas.ac.uk

*1) L45-6: as written, the sentence unintentionally I think implies these ecosystems might be restricted to the Dry Valleys. More accurately, systems containing only microarthropods and microinvertebrates are known throughout Antarctica, and in the Antarctic Peninsula/Scotia Arc actually extend up to the South Orkney and South Sandwich Islands (60 and 55-59 deg south, respectively), where there are no native true insects (two species of which do occur in parts of the Antarctic Peninsula and South Shetlands). Admittedly these communities are more diverse in species numbers and with greater biomass than those of the Dry Valleys.*

We thank Dr. Convey for catching this error. The sentence is edited/corrected on line 47.

*2) To expand the 'most extreme environment' literature beyond the Dry Valleys, there is a recent publication by Collins et al. (2020, PNAS) on part of the Transantarctic Mountains, Hodgson et al. 2010 (Antarct Sci) on the Dufek Massif (far eastern Transantarctic Mountains), and Convey & McInnes 2005 (Ecology) on nunataks in Ellworth Land, with the latter two reporting invertebrate communities in which even nematodes are absent.*

We thank Dr. Convey for these references and included them for the text on lines 50-51.

*3) L54: repeat of word 'concentrations'*

We made this correction around line 54.

*4) L59: the two Collins et al papers would be appropriate to cite here, as the most recent and detailed examples of such studies.*

We added these references on lines 61-62.

*5) L61: I would add water availability here too*

We added hydrology to line 64.

*6) L155-6: my ignorance, but what drives the choice of the proportions of samples included in the training and testing sets?*

We have updated the text to indicate that the size of the training and test datasets were based on ideal model parameters from Breiman (2001) on lines 161-162

*7) L232: is 'out of the bag' a widely used phrase?*

This was a typo and meant to say out-of-the-box. We have made this correction throughout the text.

*9) Section 5.2: is there a case that much of the text in this section might be more appropriate as a subsection in Results?*

Previous response: We agree with Dr. Convey that some of this section would typical be considered results. However, we argue that it is most logical to keep the text in the discussion given the greater context of our findings. In Section 5.2, we discuss the

strength of the model and how the strength relates to its predictive capabilities, both for the Shackleton Glacier region and other portions of the Transantarctic Mountains. We believe that this is best suited for the discussion especially since this section builds from Section 4.3, where we evaluate the model performance.

*Natasja van Gestel (Referee) natasja.van-gestel@ttu.edu*

*1) Log-transformation need to be done where necessary. First, the authors have not checked their regression models to see how the residuals show a pattern with fitted values. This is important to do as one of the (several) assumptions in a linear model are that residuals are normally distributed and their spread around the regression line should be the same irrespective of the value of x.*

*I highly suggest that the authors use log-transformed values in the data where needed and do some model checking with the plot() function and other model checking procedures. This will certainly help with that aspect. I was surprised to see that in the figure they did use log-values, but then did not use it in their regression. Needless to say, this this also needs to be done in their random forest models. The random forest model they used explained 43% of the variance in total salts. By using log-transformed values that went up to 75%. This also altered the importance ranking of the variables. Irrespective, elevation remained important (at least for Total Salts, I did not check the others), but others switched.*

> We thank Dr. van Gestel for pointing out this assumption in our statistical methods. We have log transformed our geochemical data in the regressions and have tested the normality. All regressions have a normal distribution as tested by plotting the residuals and running a Jarque-Bara Test (see script and line 150). We indicated that the data were log-transformed for analysis on line 150 and in tables 2-4. The results (especially lines 185-191 and 238-246) and discussion (especially lines 323-347) were updated accordingly.

*2) Data, such as NP, can have many zeroes. For example, the NP training data set had 116 zeroes of the 189 values. That means that a gaussian distribution of the data set is not followed. Having 0's means also a log-transformation will lead to -Inf, and cannot be used in a model. Solution: consider other family of distributions by using "glm" instead of lm. The "g" stands for generalized, and can thus handle other kinds of distributions.*

> We have chosen to simply remove the zeroes and replaced them with blank (NA) values since we are log transforming the data, as indicated on lines 151 and 161. As such, we have also added na.rm functions to our models to exclude the NA values. This is reflected in the updated script. We tested with both glm and lm and the results/strength are the same. Additionally, by log transforming the concentration data, we have achieved a normal distribution. Therefore, we have continued to use the linear model, but thank Dr. van Gestel for the suggestion on generalized linear models and will consider the model in future studies.

*3) Given that multiple samples were collected at each transect at 11 different locations, some in closer proximity than others, that error structure is not taken into account. For example, two samples from the same transect will likely be more closely related than two samples from different transect. To incorporate this, I propose using a mixed-effects model approach to take into account that multiple samples were obtained from the same transect. Otherwise: your power*

*is inflated, because your samples are not truly independent from each other (another important assumption in linear models with only fixed effects).*

> Previous response: We thank Dr. van Gestel for the suggestion on using a mixed model approach. From our understanding, a mixed model approach typically would best be applied in a situation where random plus fixed effects are not considered, such as proximity to other samples. However, we argue that by performing our regressions with geography (distance from glacier, elevation latitude, etc.), we have in fact accounted for random effects. Additionally, we have further researched mixed effects models. In summary, by performing the linear regressions, we are ultimately solving for least squares, which provides the best linear unbiased estimator. This is essentially the Gauss–Markov theorem. By solving for least squares, we have inherently proven unbiased estimators and only consider fixed effects. We thank Dr. van Gestel for helping to lead us on this informative endeavor.

*4) For the testing and training data set: rather than randomly sample all 220 observations, randomly sample a proportion of the total transects, say 8 out of 11 transect. Then test it on the remaining 3. That takes into account that the observations within a transect are not entirely independent.*

> Previous response: An important point to make here and above (#3) is that the samples are not necessarily related because of the transect sampling scheme. Any partial dependence is fundamentally driven by latitude and longitude, which are considerations in our machine learning models. This is especially important because we are testing individual points and not transects. We have also chosen to randomly sample all 220 observations to further randomize our bias and increase potential sample value variability within the datasets. As Dr. van Gestel points out, we have about 3 transects with about 8 samples each from 11 locations. For simplicity, let's assume that this corresponds to 30 transects. In order to keep the number of observation in the training and test datasets to ~85% and 15%, respectively, we would only be able to have about 4 transects in the training set. Considering that our ion concentrations span many orders of magnitude throughout the region (see Table 1), it is unlikely we are going to capture enough randomness and variance to generate strong enough models for outliers using a transect approach. This is entirely due to sample size, which we repeatedly recognize as a weakness of this study. We will certainly consider the approach proposed by Dr. van Gestel in the future when the number of observations is much greater.

*5) For PCA: this is another linear model. I found no information on whether the authors performed any visualization to see if the patterns are linear. Given that the data show non-normal distributions, please do revisit this. Also: is the PCA based on the covariance or the correlation matrix? So, information is missing.*

> We deflect to the points we have made earlier in this review regarding gaussian distributions with our log transformed data. We apologize for not specifying the PCA.

We have updated the text (line 207) and figure 4 caption to state that the PCA was based on the correlation matrix.

*6) Lastly, and I refer to Figure 2: the panels are great, but it also highlights that the authors looked at every possible relationship of the total water-soluble salts, N:P, and ClO4- and ClO3- concentrations. However: the more comparisons are made, the higher the probability of making a type 1 error, unless you make the alpha more stringent. In a scenario like this I would recommend something like a Bonferroni correction.*

We thank Dr. van Gestel for suggesting the Bonferroni Correction. We have applied the correction, which only affected the relationship between $ClO_3^-$ and elevation. This is indicated in figure 2 and lines 188-189.

*7) Minor concerns: Replace 'environmental parameters' with 'environmental variables' or 'environmental conditions'. From a model-perspective, parameters are associated with models, e.g. coefficients are parameters. Variables are the data.*

We made this correction on lines 39 and 370.

*8) L. 158 It is mean squared error.*

We made this correction on line 164.

*9) Figure 2: Please add the meaning of the blue, yellow and gray colors. It is evident from the next figure, but having it already here will help the reader. Figures are standalone and should be interpretable without having to look for info elsewhere in the paper. Also, technically: if a relationship is not significant, one should not show the best-fit line. However, rather than removing them, I would suggest adding dashed lines instead for those where P-values are greater than 0.05.*

We added the zone coloring to the figure 2 caption. We also followed Dr. van Gestel's suggestion and added dashed lines for non-significant relationships with the Bonferonni correction applied (only changes for $ClO_3^-$ vs. elevation).

*10) Supplementary information: Please add the "library(readxl)" to the R script. Otherwise users will get an error message: the read_excel() function is used, which is from that package.*

We added this line to the beginning of the script.